# Comprehensive Performance-Oriented Multi-Objective Optimization of Hemispherical Resonator Structural Parameters

**DOI:** 10.3390/mi16030287

**Published:** 2025-02-28

**Authors:** Xiaohao Liu, Xin Jin, Chaojiang Li, Yumeng Ma, Deshan Xu, Simin Guo

**Affiliations:** School of Mechanical Engineering, Beijing Institute of Technology, Beijing 100081, China; xiaohao_bit@163.com (X.L.);

**Keywords:** hemispherical resonator, multi-objective optimization, thermoelastic damping quality factor, PSO-BP, NSGA-II

## Abstract

The hemispherical resonant gyroscope is the highest-precision solid-state vibration gyroscope, widely applied in aviation, aerospace, marine, and other navigation fields. As the core component of the hemispherical resonant gyroscope, the design of its structural parameters directly influences the key performance parameters of the resonator—specifically, the thermoelastic damping quality factor and the minimum frequency difference from interference modes—affecting the operational accuracy and lifespan of the gyroscope. However, existing research, both domestic and international, has not clarified the effect of structural parameters on performance laws. Thus, studying the mapping relationship between the resonator’s performance and structural parameters is essential for optimization. In this study, a hemispherical resonator with a midplane radius of 10 mm serves as the research object. Based on a high-precision finite element simulation model of an ideal hemispherical resonator, the mechanism of thermoelastic damping and the influence of structural parameters on performance are analyzed. A PSO-BP neural network mapping model is then developed to relate the resonator’s structural and performance parameters. Subsequently, the NSGA-II algorithm is applied to perform multi-objective mapping of these parameters, achieving an optimized resonator with a 4.61% increase in the minimum frequency difference from interference modes and a substantial improvement in thermoelastic damping of approximately 70.41%. The comprehensive, performance-oriented multi-objective optimization method for the structural parameters of hemispherical resonators proposed in this paper offers a cost-effective approach to high-performance design and optimization, and it can also be applied to other manufacturing processes under specific conditions.

## 1. Introduction

The hemispherical resonant gyroscope (HRG) is the highest-precision solid-state vibration gyroscope. Compared with traditional gyroscopes—such as mechanical, fibre-optic, and laser gyroscopes—the HRG offers advantages such as measurement accuracy, compact size, low power consumption, extended lifespan, broad measurement range, and reduced preparation time, making it a “game-changer” in inertial technology [1,2,3,4]. As the HRG’s core component, the resonator is subject to stringent export controls by other countries. While the diameter of the resonator is available, details on the central rod diameter, rod length, and spherical shell thickness remain undisclosed. Thus, optimizing the resonator’s structural parameters has become a key research focus in HRG technology [5,6,7]. In addition, vibration frequency is a crucial design parameter for hemispherical resonators. Achieving a high accuracy of the HRG requires a long time for the vibrations to decay in the resonator. For this, the resonator’s quality factor must be as high as possible, and the vibration frequency must be low. However, to reduce the vibration frequency, it is necessary to reduce the thickness of the resonator wall, which complicates the production of resonators and can disrupt the axial symmetry of the resonator. That is why the vibration frequency of aero-space-grade HRGs typically ranges from 4 to 8 kHz.

In designing the hemispherical resonator, the quality factor and minimum frequency difference from interference modes are core performance parameters. The quality factor serves as a key index to measure the HRG’s energy loss, with a high-quality factor indicating low energy loss, prolonged free decay time, and strong suppression of interference frequencies outside the operational frequency, thereby enhancing the HRG’s accuracy [8]. The minimum frequency difference represents the smallest frequency gap between the resonator’s operational modes and the adjacent modes. In this paper, this parameter is referred to as the “minimum frequency difference”. A larger minimum frequency difference helps prevent coupling between the working and adjacent modes, thereby ensuring stable resonator operation and an improved HRG accuracy [9,10]. During the HRG operation, periodic deformation of the hemispherical resonator due to vibration induces a temperature gradient, creating an irreversible heat flow and resulting in energy loss, known as thermoelastic damping (TED). Thermoelastic loss is a primary form of energy dissipation in the resonator, determining the resonator’s thermoelastic damping quality factor (*Q_TED_*) [11,12,13,14,15].

To uncover the generation mechanism of thermoelastic damping, scholars worldwide have conducted extensive research. In the 1930s, Zener [16,17] proposed a model for calculating the thermoelastic damping quality factor for wires and spring plates, taking into account the one-dimensional heat transfer in their bending direction. Later, Duwel [18] and colleagues introduced a “weakly coupled” method into Zener’s thermodynamic model, weakly coupling thermodynamic and mechanical dynamics equations to calculate the thermoelastic damping quality factor for resonators of different structures, significantly expanding Zener’s theory. Didace [19] developed a thermo-electro-mechanical FEM-BEM model to simulate the MEMS resonators, enabling an in-depth analysis of their energy dissipation mechanisms. Sorenson et al. [20] developed a finite element model to simulate thermoelastic damping in micro-hemispherical resonators. Their analysis examined damping generation across the resonator’s body and surface, finding that under the Rayleigh non-tensor assumption, perturbations in the shell’s neutral surface contribute to thermoelastic damping. Darvishian et al. [21] developed an analytical model to predict the thermoelastic damping quality factor, studying how material properties, shell geometry, edge chipping, trimming methods, film coatings, and operating temperature affect this factor. They proposed enhancing the damping quality factor by optimizing edge trimming techniques. Sharma et al. [22,23] investigated the effects of structural shape, dimensions, materials, and coatings on the resonator’s quality factor. They found that energy dissipation in thin-film metal coatings increases significantly due to thermoelastic damping and internal friction within the coating material. Additionally, anchoring damping was found to be highly sensitive to fabrication defects. This study supports the development of a comprehensive strategy for designing, realizing, and operating mechanical resonators. Khooshehmehri et al. [24] developed a finite element model of a hemispherical shell resonator, performing modal analysis and thermal simulations to examine how resonator geometry, structural parameters, and material properties affect thermoelastic and anchorage damping. This work provides useful guidance for structural design in hemispherical shell resonators. Xiaoyan Sun et al. [25] enhanced the thermoelastic quality factor by adjusting the heat conduction distance, adding thermal insulation cavities and layers at the resonator’s top and edges, leading to a 13.8-fold improvement in the quality factor, compared to the original structure. Ma Chuanzhen et al. [26] analyzed the effects of structural parameters, machining errors, and surface quality on the quality factor of thermoelastic damping, though only a single-factor analysis was performed. This approach, however, does not fully address the coupled effects of different structural parameters, limiting its guidance for precise hemispherical resonator design. In summary, research on thermoelastic damping in hemispherical resonators has primarily focused on how material properties, geometric structure, surface coatings, and processing errors impact the quality factor. Few studies, however, have explored the influence of structural parameters on the thermoelastic damping quality factor from a fabrication perspective. Furthermore, optimal structural design for hemispherical resonators requires a mapping model that links performance parameters, such as the thermoelastic damping quality factor and minimum frequency difference, to structural parameters. Such a model would facilitate multi-objective optimization in resonator design.

To address the limitations identified in previous studies, this paper focuses on a fused silica hemispherical resonator with a midplane radius of 10 mm. A high-precision finite element simulation model is established to analyze the mechanism of thermoelastic damping and the impact of structural parameters on performance characteristics, culminating in a mapping model from the resonator’s structural parameters to its performance parameters. Using the NSGA-II algorithm, this study achieves multi-objective optimization of the hemispherical resonator’s structural parameters, aiming for improved thermoelastic damping and a larger minimum frequency difference as optimization objectives.

## 2. Modelling and Analysis

### 2.1. Modeling of the Ideal Hemispherical Resonator for Finite Element Simulation

This paper analyzes the impact of the “Ψ” hemispherical resonator’s geometric parameters on its thermoelastic damping and resonance frequency, addressing a thermodynamic coupling problem. Therefore, a finite element simulation model of the hemispherical resonator was developed using the multi-physics coupling software, COMSOL Multiphysics 6.0, in the Solid Mechanics and Heat Transfer in Solids modules. A schematic representation of the hemispherical resonator’s structure is shown in Figure 1a,b.

In the Solid Mechanics module, the upper and lower ends of the central rod serve as the clamping and fixed ends, respectively, where the fixed constraints are applied to simulate actual working conditions. The upper and lower surfaces also act as temperature boundaries, with the initial temperature set to 293.15 K.

In finite element analysis, mesh partitioning significantly influences the solution results. Given the current expectation for high-quality hemispherical resonators to crack at frequency levels in the millihertz range, the frequency difference attributable to the simulation model must be less than 0.0001 Hz. However, conventional mesh partitioning methods currently yield frequency differences between 0.03 and 0.1 Hz, which do not meet the analysis requirements. The model error associated with the mesh primarily arises from the lack of strict symmetry of the grids around the axis of the central pole. To address this issue, a multi-zone method is proposed for meshing the resonator. This method divides the resonator into four equal parts, first generating a uniform quadrilateral mesh in the two-dimensional plane, then rotating the mesh around the axis of the central pole and then finally creating a uniform sweeping mesh for each section. The mesh subdivision is illustrated in Figure 1c.

The initial structural parameters of the hemispherical resonator examined in this paper are presented in Table 1. The material used for the hemispherical resonator is fused silica, and its mechanical and thermodynamic performance parameters are provided in Table 2.

A modal analysis of the hemispherical resonant gyro is conducted to calculate the vibration frequencies and modes associated with each operational mode of the device. Based on the working principle of the hemispherical resonant gyro, its operational mode is the two-wave belly vibration mode when the modal vibration order *n* = 2, with the fourth and fifth orders representing the active modes at this state. When the hemispherical resonator exhibits asymmetric errors (such as an uneven mesh division), the original two-wave belly vibration mode splits into two orthogonal modes. The principal axis of these modes rotates by 45° relative to the original *n* = 2 mode, transforming into a four-wave belly vibration (*n* = 4). Consequently, the intrinsic frequencies of these two modes, originally identical, develop a certain difference [27,28]. The magnitude of this frequency difference, known as frequency splitting, depends on the degree of asymmetry in the resonator structure. Frequency splitting, resulting from meshing errors, pertains to the fourth and fifth order modes. These discrepancies cause the angle of the intrinsic rigidity axis to be 45°, with a frequency crack value of 0.000099 Hz, thereby satisfying the requirements for high-precision simulation.

Different meshing methods directly influence the results of the modal analysis of the hemispherical resonator, with the corresponding frequency discrepancies for each method presented in Table 3. The multi-zone method employed in this paper, in conjunction with the resonator area division method, significantly enhances the accuracy of the resonator simulation model, thereby establishing a foundation for subsequent simulation analyses of the thermoelastic damping of the hemispherical resonator.

### 2.2. Analysis and Validation of Finite Element Model

Under the operating mode of four-wave belly vibration, the hemispherical shell periodically experiences tension and compression. The temperature of the regions subjected to compressive stress increases, while the temperature of those under tensile stress decreases. The temperature field changes, resulting due to these stresses, are illustrated in Figure 2a. At this moment, the temperature gradient is maximized along the lip of the hemispherical shell and the central rod, particularly at the rounded corners where they connect. Heat flows from the high-temperature region to the low-temperature region, resulting in irreversible heat loss of the vibration energy of the hemispherical resonator. This energy loss is referred to as thermoelastic damping loss. There are five primary paths of heat flow resulting from the temperature gradient, as illustrated in Figure 2b.

1. In the direction of the wall thickness of the hemispherical shell.

2. In the direction of the circumference of the hemispherical shell.

3. In the direction of the wall thickness of the transition fillet.

4. In the direction along the circumference of the transition fillet.

5. In the direction from the lip to the transition fillet along the hemispherical shell’s bushing.

For the analysis of the elastic strain energy in the working mode of the hemispherical resonator, the energy is primarily concentrated in the transition fillet and the lip region of the spherical shell, as illustrated in Figure 3.

Therefore, based on the modular calculation method for *Q_TED_* [21], the *Q_TED_* of the hemispherical resonator can be calculated as follows:(1)$$\frac{E_{tot}}{Q_{tot}}=\frac{E_{r}}{Q_{r}}+\frac{E_{R}}{Q_{R}}$$
where *E_tot_*, *E_r_*, and *E_R_* represent the total elastic strain energy of the hemispherical resonator, the elastic strain energies of the transition fillet, and the lip regions of the hemispherical shell, respectively. While *Q_tot_*, *Q_r_*, and *Q_R_* denote the thermoelastic damping factors for the overall structure, the transition fillet, and the lip region. According to the results of the finite element calculation, we can obtain *E_r_* = 0.645*E_tot_*, *E_R_* = 0.355*E_tot_*, thus (1) can be simplified as:(2)$$\frac{1}{Q_{tot}}=\frac{0.645}{Q_{r}}+\frac{0.355}{Q_{R}}$$

The thermoelastic damping in the lip region of the hemispherical resonator can be calculated using the thermoelastic damping quality factor model for simply supported beam resonators proposed by Zener:(3)$$Q_{R}=\frac{\rho C}{E\alpha^{2}T_{0}}\frac{1+{\left(\omega \cdot \tau_{across-t}\right)}^{2}}{\omega \cdot \tau_{across-t}}$$
where *ρ*, *E*, *α*, *C*, and *T*_0_ represent the density, Young’s modulus, coefficient of thermal expansion, constant pressure heat capacity, and initial temperature of the fused silica material, respectively. *ω* is the angular frequency of the hemispherical resonator’s vibration, and *τ*_across-t_ is the heat transfer time constant for heat flow through the wall thickness of the spherical shell, which can be calculated using the following formula:(4)$$\tau_{across-t}=\frac{t^{2}}{\pi^{2}D}=\frac{t^{2}\rho C}{\pi^{2}\kappa }$$
where *D* and *κ* represent the thermal diffusion coefficient and thermal conductivity of the material, respectively, and *t* denotes the length of the heat flow path, which is the wall thickness of the hemispherical shell.

The thermoelastic damping *Q_r_* in the transition fillet region comprises the heat flow through the wall thickness of the transition fillet and the heat flow along its circumference, calculated using the following formula:(5)$$\frac{1}{Q_{r}}=\frac{E\alpha^{2}T_{0}}{\rho C}\frac{\omega \cdot \tau_{\mathrm{across}-z}}{1+{\left(\omega \cdot \tau_{\mathrm{across}-z}\right)}^{2}}+\frac{E\alpha^{2}T_{0}}{\rho C}\frac{\omega \cdot \tau_{\mathrm{along}-r}}{1+{\left(\omega \cdot \tau_{\mathrm{along}-r}\right)}^{2}}$$
where *τ*_across-z_ and *τ*_along-r_ are the heat transfer constants for the heat flow through the wall thickness of the transition fillet and along its circumference, respectively, which can be approximated using the following formula:(6)$$\tau_{\mathrm{across}-z}=\frac{{\bar{z}}^{2}}{\pi^{2}D}=\frac{{\bar{z}}^{2}\rho C}{\pi^{2}\kappa }$$(7)$$\tau_{\mathrm{along}-r}=\frac{{\left(2\pi a\right)}^{2}}{\pi^{2}D}=\frac{4\rho Ca^{2}}{\kappa }$$
where $\bar{z}$ is the average heat path length for the heat flow through the wall thickness of the transition fillet, and a is the radius of the central rod.

The heat flow along the circumference of the hemispherical resonant shell and from the lip to the transition fillet heat transfer path is extensive. The time required for this heat transfer is significantly longer than the vibration period of the second-order resonant state of the hemispherical resonator, indicating that there is insufficient time for heat transfer. Consequently, this paper treats these two heat flow processes as adiabatic and ignores them.

By substituting the geometrical parameters from Table 1 and the material parameters from Table 2 into (1)–(7), the thermoelastic damping of the ideal hemispherical resonator is calculated to be 1.3254 × 10^9^. The result from the established finite element model is 1.247 × 10^9^. The simulation model and the theoretical model for *Q_TED_* of the ideal hemispherical resonator show good agreement in determining the quality factor.

Additionally, referencing the research model from the literature [18], this paper’s simulation model is utilized to calculate the influence of the midplane radius and shell thickness of the hemispherical resonator on *Q_TED_*, and to compare these results. As shown in Figure 4, the horizontal axis represents the rate of change in the actual midplane radius of the resonator relative to its standard value, while the vertical axis denotes the rate of change in the actual thermoelastic damping factor *Q_TED_* of the resonator relative to the standard parameter *Q_TED_*. The results indicate that the findings of this paper align closely with those in the literature. Therefore, it can be concluded that the simulation model presented in this paper is reliable.

### 2.3. Analysis of the Effect of Structural Parameters of Hemispherical Resonators on Vibration Frequency and Thermoelastic Damping

To investigate the influence of the structural parameters of the hemispherical resonator on vibration frequency and thermoelastic damping, this study focuses on the midplane radius *R*, spherical shell thickness *t*, central rod radius *a*, length of the clamped end of the central rod *L*_1_, length of the fixed end of the central rod *L*_2_, and both the inner and outer fillet radii *r*. The effects of these parameters on the operating mode frequency *f*, minimum frequency difference in the interference mode ∆*f_min_*, and thermoelastic damping *Q_TED_* are simulated using the established finite element model. The specific parameter settings are detailed in Table 4.

Figure 5 illustrates the effect of the midplane radius of the resonator *R* on the operating mode frequency *f*, minimum frequency difference in the interference modes ∆*f_min_*, and thermoelastic damping *Q_TED_* of the hemispherical resonator at various spherical shell thicknesses *t*. *f* of the hemispherical resonator decreases as *R* increases. For *t* between 0.5 mm and 1.25 mm, ∆*f_min_* decreases monotonically with the increase in R. However, when *t* was beyond 1.25 mm, ∆*f_min_* was within a range of 1000 Hz as the midplane radius *R* increases, indicating a relatively poor anti-interference capability of the hemispherical resonator. In addition, when the ratio of *t* to a exceeds 1/3, the working modes are found in the 4th and 5th orders. Conversely, if this ratio is less than 1/3, the working modes shift to the 1st and 2nd orders, facilitating the stimulation of low-order modes and effectively enhancing the sensitivity characteristics of the hemispherical resonance gyroscope. *Q_TED_* decreases significantly with the increase in the midplane radius in the range of 9–15 mm. In contrast, for *R* between 16 mm and 19 mm, the *Q_TED_* tends to be stabilized as the midplane radius increases.

Figure 6 illustrates the influence of the spherical shell thickness *t* on the operating mode frequency *f*, the minimum frequency difference ∆*f_min_*, and the thermoelastic damping *Q_TED_* of the hemispherical resonator at different midplane radii *R*. *f* increases linearly with the *t*. ∆*f_min_* associated with the interference mode exhibits a ‘W’ distribution as *t* varies from 0.5 mm to 1.5 mm. Specifically, it monotonically decreases in the range of 0.75 mm to 0.7 mm, and again from 0.75 mm to 1.5 mm. Within the range of 0.5 mm to 0.7 mm, ∆*f_min_* decreases monotonically; from 0.75 mm to 1.0 mm, it increases monotonically; from 1.05 mm to 1.3 mm, it decreases monotonically; and from 1.35 mm to 1.5 mm, it increases again. When *t* = 0.5 mm or *t* = 1 mm, ∆*f_min_* from the interference mode reaches a larger value. *Q_TED_* also increases linearly with *t*. This occurs because the increasing *t* extends the heat transfer path in the direction of heat flow, resulting in a longer time constant for heat transfer. This reduces energy loss during the vibration cycle, thereby enhancing *Q_TED_*.

Figure 7 illustrates the influence of the fillet radius *r* on the operating mode frequency *f*, the minimum frequency difference in the interference modes ∆*f_min_*, and the thermoelastic damping *Q_TED_* of the hemispherical resonator with varying radii of the centre rod *a*. *f* increases monotonically with *r*. However, as a varies, ∆*f_min_* exhibits a different trend with increasing *r*. When *a* = 2.5 mm, ∆*f_min_* decreases as *r* increases. However, at this point, all values of ∆*f_min_* are below 1000 Hz, indicating poor interference resistance of the hemispherical resonator. When *a* = 3 mm, ∆*f_min_* first increases and then decreases and then increases with the increase in r, and when *r* = 1 mm, ∆*f_min_* obtained the maximum value. When *a* is in the range of 4~5 mm, ∆*f_min_* increases and then decreases with the increase in r, and ∆*f_min_* achieves the maximum value near *r* = 2 mm. *Q_TED_* exhibits fluctuating changes with increasing *r*, though the magnitude of these changes is relatively small.

Figure 8 illustrates the influence of the centre rod radius *a* on the operating mode frequency *f*, the minimum frequency difference from the interference mode ∆*f_min_*, and the thermoelastic damping *Q_TED_* of the hemispherical resonator with varying radii *r*. *f* increases linearly with *a*. When *r* is in the range of 1.6 mm to 2 mm, ∆*f_min_* initially increases with *a*, then decreases, and subsequently increases again, reaching a local maximum in the range of 2.8 mm to 3 mm. *Q_TED_* exhibits fluctuating changes with increasing *a*, though the magnitude of these changes is relatively small.

Figure 9 illustrates the influence of the clamping end length of the centre rod *L*_1_ on the operating mode frequency *f*, the minimum frequency difference from the interference mode ∆*f_min_*, and the thermoelastic damping of the hemispherical resonator *Q_TED_* with varying lengths of the fixed end of the centre rod *L*_2_. When the fixed end constraint length is constant, *L*_2_ has no effect on *f*, ∆*f_min_*, or *Q_TED_*. *f* increases with *L*_1_ and then tends to stabilize. ∆*f_min_* increases monotonically with *L*_1_, while *Q_TED_* fluctuates with increasing *L*_1_, exhibiting *a* relatively small change of approximately 17.53%.

In summary, while the graphs reflect the relationship between individual structural parameters and performance parameters to some extent, interpreting this mapping is challenging due to the interactions among the structural parameters. Therefore, a global optimization of all structural parameters is necessary.

## 3. Mapping Model of Performance Parameters and Structural Parameters for the Hemispherical Resonator

Based on the finite element model and the structural parameters presented in Table 4, 550 sets of data were calculated. In this paper, a multivariate linear regression model, a multivariate nonlinear regression model, and a BP neural network model, optimized using a particle swarm algorithm, are sequentially established to analyze the relationships between the three performance parameters and the six structural parameters. The prediction effects of these three models are then compared.

### 3.1. Multiple Linear Regression Model

Based on the principle of least squares, MATLAB R2022b software was utilized to conduct multiple linear regression analysis on six structural parameters and three performance parameters (*f*, ∆*f_min_*, and *Q_TED_*) of the hemispherical resonator. Eighty percent of the data were used as the training set, while the remaining twenty percent served as the test set. The resulting multiple linear regression models are presented in (8)–(10):(8)$$\begin{array}{l} f=17608.58-1257.06x_{1}+4962.635x_{2}\\ +261.0931x_{3}+820.7439x_{4}+121.2194x_{5}-182.825x_{6} \end{array}$$(9)$$\begin{array}{l} \Delta f_{\min}=1835.95-150.75x_{1}-801.19x_{2}\\ +444.527x_{3}+244.328x_{4}+8.01647x_{5}+41.3473x_{6} \end{array}$$(10)$$\begin{array}{l} Q_{TED}=3\times 10^{8}-6.1\times 10^{7}x_{1}+2.04\times 10^{9}x_{2}\\ -1.4\times 10^{8}x_{3}+4.383\times 10^{7}x_{4}-2.457\times 10^{6}x_{5}-2\times 10^{7}x_{6} \end{array}$$
where *x*_1_–*x*_6_ denotes the six structural parameters of the hemispherical resonator: midplane radius *R*, spherical shell thickness *t*, centre rod radius *a*, inner and outer fillet radii *r*, centre rod clamping end length *L*_1_, and centre rod fixed end length *L*_2_, in that order.

The analysis results of the multiple linear regression model are presented in Table 5. Although the goodness of fit for the regression model concerning operating frequency and thermoelastic damping *R*^2^ is greater than 0.8, the fitting results remain unsatisfactory. Furthermore, the fitting performance of the regression model for ∆*f_min_* is even worse, indicating that the overall fitting of the multivariate linear regression model is inadequate.

### 3.2. Multivariate Nonlinear Regression Model

To improve the model fitting accuracy, the structural parameters and performance parameters were analyzed using multivariate nonlinear regression based on the least squares method. After several tests, a 6th full-order polynomial was selected as the multivariate nonlinear regression model, which includes all terms from the 1st to the 6th order of the independent variables, as well as all possible interaction terms. The analysis results of the multivariate nonlinear regression model are presented in Table 6 and Figure 10, Figure 11 and Figure 12. The goodness of fit *R*^2^ of the multivariate nonlinear regression model for *f* and *Q_TED_* is greater than 0.97, indicating an excellent fit and making the model suitable for data prediction. Although the goodness of fit for the multivariate nonlinear regression model concerning ∆*f_min_* has significantly improved, it remains below 0.9. Therefore, it is necessary to explore additional models to enhance fitting accuracy.

### 3.3. PSO-BP Neural Network Model

The BP neural network is one of the most widely used neural network models and is classified as a multilayer feedforward neural network. The structure of the BP neural network developed in this paper is illustrated in Figure 13. This unidirectional multilayer feedforward network comprises an input layer, a hidden layer, and an output layer. Each layer consists of multiple neuron nodes, interconnected by weight values. The learning process involves forward propagation of the signal and backward propagation of the error. The fundamental principle is that in a feedforward network, the input signal is processed through the input layer, and the output layer computes the output via the hidden layer, comparing the output value with the desired value. If an error occurs, it propagates backward from the output layer to the input layer. During this process, the neuron weights are adjusted using the gradient descent algorithm. The BP neural network allows us to derive the mapping relationship between multiple structural parameters and performance variables.

However, the BP neural network algorithm has certain defects and shortcomings; it converges slowly, requiring hundreds or even thousands of iterations to solve even simple problems. Furthermore, the traditional BP neural network algorithm adjusts the connection weights based on gradient information, making it susceptible to becoming trapped in local extrema. Additionally, due to the sensitivity of BP neural networks to initial weights, varying initial weights may lead the network to converge to different local minima. The selection of initial weights and thresholds for BP neural networks lacks a unified theoretical framework and is often based on empirical methods, which undermines the scientific validity of the predictions for the hemispherical resonator performance parameters.

The Particle Swarm Optimization (PSO) algorithm partially addresses the issues associated with traditional BP neural networks. We assume that the number of neurons in the hidden layer of the BP neural network, the learning rate, the initial weights, and the bias constitute the solution aggregates to be optimized. All solution aggregates are treated collectively within the framework of PSO, while each individual solution aggregate is considered separately. By combining local and global optimal solutions, this approach helps avoid convergence to local optima, allowing for the determination of unique values for the number of neurons, learning rate, initial weights, and bias of the hidden layer.

The workflow of the PSO-BP neural network algorithm developed in this paper is as follows:

(1) Data Preprocessing: A sample set is constructed from the simulation data derived from the finite element model. The input dimension consists of six parameters: midplane radius *R*, spherical shell thickness *t*, centre rod radius *a*, inner and outer fillet radii *r*, centre rod clamping end length *L*_1_, and centre rod fixed end length *L*_2_. Each output includes operating frequency *f*, minimum frequency difference from the interference modes ∆*f_min_*, and thermoelastic damping *Q_TED_*. A total of 550 data sets were collected, with 80% allocated to the training set and the remaining 20% to the test set. The input and output matrices are represented in the following equations:(11)$$X={\left[R\;\;t\;\;a\;\;r\;\;L_{1}\;\;L_{2}\right]}^{T}$$(12)$$Y={\left[f\;\;\Delta f_{\min}\;\;Q_{TED}\right]}^{T}$$

The raw data were normalized according to the following equation:(13)$$X_{i}=\frac{x_{i}-x_{\min}}{x_{\max}-x_{\min}}\left(i=1,2,\ldots \ldots ,550\right)$$
where the *X_i_* is the normalized data, *x_i_* is the original data, *x_max_* and *x_min_* are the maximum and minimum values corresponding to the original data set for each parameter, respectively.

(2) Training Parameter Settings: Prior to training the PSO-BP neural network, it is necessary to configure the training parameters. Specifically, the maximum number of iterations for the training is set to 1000, with a required accuracy of 1 × 10^−6^, while default values are utilized for other parameters. These training parameters are established to ensure that the network achieves the desired training effect within a reasonable timeframe, thereby avoiding excessively long training durations or infinite loops.

(3) Particle Swarm Algorithm Optimization: In the first stage, the number of neurons in the hidden layer of the BP neural network and the learning rate are designated as the target parameters for optimization via the particle swarm algorithm. In the second stage, the initial weights and bias of the BP neural network serve as the target parameters for optimization. The specific implementation steps are as follows:

➀ Initialize the PSO Parameters: Set the number of population updates *F* = 30, population size *p* = 10, individual learning factor *c*_1_ = *c*_2_ = 2, and population learning factor. Initialize the particle swarm position and velocity, constraining them to the range of (−1, 1). The fitness function is defined as the mean square error of network predictions, aimed at minimizing network error. ➁ First Stage PSO: Calculate the initial particle fitness based on the fitness function to identify the initial optimal particles. Subsequently, compute the population fitness and update the speed and position of the particles. This process determines the history of individual optimal solutions and global optimal solutions. After several iterations, all particles in the swarm converge towards the optimal solution, achieving either the minimum error accuracy or the maximum number of iterations. This stage outputs the optimal number of neurons in the hidden layer and the learning rate. ➂ Second Stage PSO: The optimal number of hidden layer neurons and the learning rate are incorporated into the parameters of the BP neural network structure. This step repeats the process outlined in step 2, resulting in the output of the optimal initial weights and bias. ➃ Execute the BP Neural Network Post-PSO: The optimal number of hidden layer neurons, learning rate, initial weights, and bias are integrated into the BP neural network structure, followed by running the neural network to derive the optimal training model.

The specific PSO-BP neural network algorithm flowchart is shown in Figure 14.

After iterative calculations, the optimized number of hidden layer neurons is determined to be 13, with a learning rate of 0.1. The change curve of the optimal fitness for the particle swarm is illustrated in Figure 15. It can be observed that throughout the iteration cycles, the particle swarm gradually converges towards optimal fitness. Around 15 iterations, the optimal fitness stabilizes, indicating that the particle swarm algorithm is highly efficient in the optimization process and enables rapid convergence to the optimal solution.

The prediction results for the performance parameters of the hemispherical resonator using the PSO-BP model are presented in Figure 16, Figure 17 and Figure 18, with the corresponding evaluation indices provided in Table 7. The goodness of fit *R*^2^ for the PSO-BP neural network model concerning the operating mode frequency *f*, minimum frequency difference with the interference mode ∆*f_min_*, and thermoelastic damping *Q_TED_* all exceeds 0.9, outperforming the multivariate nonlinear model. Regarding root mean square error, the PSO-BP neural network model demonstrates improvements in prediction accuracy of 36.4%, 25.4%, and 18.6% over the original BP neural network model, respectively. In terms of mean absolute error, the PSO-BP neural network model shows improvements in prediction accuracy of 36.3%, 22.1%, and 20.7% over the original BP neural network model. This indicates that the particle swarm algorithm enhances the model’s prediction accuracy by optimizing the initial weights and biases of the BP neural network. This model establishes a foundation for the multi-objective optimization of the structural parameters of the hemispherical resonator.

## 4. Multi-Objective Optimization of the Structural Parameters of the Hemispherical Resonator Using the NSGA-II Algorithm

NSGA-II is a genetic algorithm, frequently employed to address combinatorial optimization problems. It utilizes elite strategies and crowding distance concepts, enhancing the retention probability of superior individuals while reducing global complexity. The PSO-BP neural network trained in this study is initially designated as the constraint function for the NSGA-II algorithm. Subsequently, the optimal combinations of structural parameters corresponding to the best performance parameters, constrained by a specific manufacturing process, will serve as the output of the NSGA-II.

Prior to employing the multi-objective optimization algorithm, the range of optimization variables is established, and the constraints of the multi-objective optimization algorithm are presented in the following equation, as outlined in Table 4. In addition, the vibration frequency should be constrained within the range of 4–8 kHz.(14)$$\left\{\begin{array}{l} 9\leq R\leq 19\\ 0.5\leq t\leq 1.5\\ 2.5\leq a\leq 5\\ 1\leq r\leq 3\\ 0\leq L_{1}\leq 5\\ 3\leq L_{2}\leq 10\\ 4000\leq f\leq 8000 \end{array}\right.$$

The optimization objectives of this study are to achieve higher thermoelastic damping and a larger minimum frequency difference from the disturbed mode. Thus, the optimization model is presented in (15).(15)$$goal\left\{\begin{array}{c} \min(f_{1})=\min(-Q_{TED})\\ \min(f_{2})=\min(-\Delta f_{\min}) \end{array}\right.$$

### 4.1. Establishment of NSGA-II Multi-Objective Optimization

The flow of the NSGA-II algorithm is illustrated in Figure 19. The specific operations are as follows:

(1) Initialize the parameters of the genetic algorithm: Set the population size *Q* to 100, crossover probability *p*_cross_ to 0.8, mutation probability *p*_mutation_ to 0.05, and the number of evolutionary iterations *i*_max_ to 50.

(2) Population initialization and chromosome coding: Since the established PSO-BP model normalizes the data, the range of selected structural parameters is [0, 1]. The precision is set to 10^−2^, and the coding length of each process parameter is determined to be 7. Binary coding is employed to generate the initial population.

(3) Calculation of fitness: Different structural parameter groups are obtained by decoding the individuals in the initial population. These parameter groups are input into the PSO-BP prediction model to compute the fitness function value for each individual.

(4) Genetic manipulation of the population: Selection, crossover, and mutation are applied to create a new generation of the population. The fitness values of individuals are calculated similarly. Based on these values, individuals are ranked to guide the population toward evolution in the direction of those with lower fitness until the termination conditions for iteration are met. Finally, the individual with the smallest fitness function value is selected. This individual is decoded, and inverse normalization is applied according to the constraints of the decoded values, resulting in the Pareto optimal solution set.

### 4.2. Optimization Results and Validation

The obtained Pareto solution set is illustrated in Figure 20. The optimization process yielded 100 sets of non-inferior solutions. Based on the optimization objectives, we require a higher quality factor and a larger minimum frequency difference from the interfering modes. Empirically, when the minimum frequency difference exceeds 2000 Hz, the four-wave belly vibration of the hemispherical harmonic oscillator can effectively avoid interference from environmental factors and neighbouring modes. In addition, the vibration frequency of the optimized hemispherical resonator must be maintained within a reasonable range. Consequently, we select the 85th solution from the non-inferior set. At this point, the thermoelastic damping predicted by the PSO-BP model is 2.125 × 10^9^, the minimum frequency difference with the interfering modes is 2088.546 Hz, and the vibration frequency is measured at 7892.428 Hz. This corresponds to the following structural parameters: the midplane radius of the hemispherical shell is 11.673 mm, the wall thickness is 1.058 mm, the centre rod radius is 2.836 mm, the inner and outer fillet radii are both 1.689 mm, the length of the clamping end of the centre rod is 4.684 mm, and the length of the fixed end of the centre rod is 6.652 mm.

The optimized structural parameters were input into the finite element model for calculation, and the comparison results of the performance parameters under these structural parameters versus those before optimization are presented in Table 8. As shown in the table, the errors between the predicted and actual values of the PSO-BP neural network are below 7.31%, indicating a strong predictive accuracy. Furthermore, for the optimized hemispherical resonator, ∆*f_min_* improved by 4.61%, while *Q_TED_* achieves an even greater improvement of approximately 70.41%. This improvement is primarily due to the increased thickness of the spherical shell, which extends the heat transfer path along its thickness direction, increasing the heat transfer time constant. Consequently, this reduces energy loss during the vibration cycle, thereby enhancing thermoelastic damping. This underscores that the NSGA-II algorithm yields an optimal structural parameter scheme aimed at maximizing both ∆*f_min_* and *Q_TED_* the thermoelastic damping. However, despite the significant improvement in thermoelastic damping, subsequent machining and assembly processes still lead to a sharp decline in the quality factor.

## 5. Conclusions

This paper first establishes a high-precision finite element simulation model of an ideal hemispherical resonator. The generation mechanism of thermoelastic damping is analyzed to verify the accuracy of the finite element model. Subsequently, a data set is obtained through finite element computations that capture the performance parameters of the hemispherical resonator as the structural parameters vary, followed by an analysis of the influence of these structural parameters on the performance parameters. A two-stage optimization of the BP neural network was then performed using the elite particle swarm algorithm, resulting in the establishment of a PSO-BP neural network model for the hemispherical resonator, incorporating both structural and performance parameters. Finally, the structural parameters of the hemispherical resonator were multi-objectively optimized by employing the trained neural network as input to the NSGA-II algorithm, targeting higher thermoelastic damping and a larger minimum frequency difference from the interference modes as optimization objectives. The main conclusions are as follows:

(1) This paper employs a uniform mesh division method with a multi-zone approach to establish a high-precision finite element simulation model of the ideal hemispherical resonator. The frequency cracking value resulting from mesh errors is below mHz. Additionally, the paper analyzes the generation mechanism of thermoelastic damping and presents a theoretical calculation model to validate the accuracy of the finite element model, thereby laying the groundwork for subsequent simulations and analyses of the thermoelastic damping in hemispherical resonators.

(2) This paper establishes a mapping model between the structural and performance parameters of the hemispherical resonator. While the multiple nonlinear regression model can accurately predict the operating frequency and thermoelastic damping, it struggles to predict the minimum frequency difference with the interference modes. In contrast, the PSO-BP neural network models demonstrate goodness-of-fit values of 0.99798, 0.96777, and 0.98215 for the operating frequency, minimum frequency difference, and thermoelastic damping, respectively, indicating a high prediction accuracy and effectiveness in predicting the performance parameters of the hemispherical resonator.

(3) Utilizing the mapping model established above, the structural parameters of the hemispherical resonator were optimized through the NSGA-II algorithm for multi-objective optimization. The optimized resonator exhibits enhanced thermoelastic damping and a larger minimum frequency difference from interference modes. The optimized structural parameters are as follows: the midplane radius of the hemispherical shell is 11.673 mm, the wall thickness is 1.058 mm, the centre rod radius is 2.836 mm, the inner and outer fillet radii are 1.689 mm, the clamping end length of the centre rod is 4.684 mm, and the fixed end length is 6.652 mm. Compared to the pre-optimization configuration, the minimum frequency difference from interference modes has improved by 4.61%, while the thermoelastic damping has seen an impressive increase of approximately 70.41%. Furthermore, the structural optimization method for the hemispherical resonator, based on the PSO-BP and NSGA-II algorithms proposed in this study, mitigates waste associated with experimental machining under specific conditions and is applicable to other manufacturing processes.

However, this study has several limitations. First, the weights of the two optimization objectives—thermoelastic damping quality factor and the minimum frequency difference from interference modes—should be explicitly quantified to facilitate a more precise selection of the optimal structural parameters from the Pareto solution set. Second, while thermoelastic damping quality factor significantly influences the overall quality factor of a micro-hemispherical resonator with a smaller midplane radius and thinner shell thickness, its effect is relatively less pronounced in the 10 mm-scale hemispherical resonator investigated in this study. Therefore, the proposed methodology may be particularly beneficial for the structural design of micro-hemispherical resonators, which will be a key focus of our future research. Third, the hemispherical resonator analyzed in this study represents an idealized structure. In practical applications, fabrication errors can significantly impact its performance parameters. Thus, further investigation is required to assess the effects of manufacturing deviations on resonator performance. Finally, in our future work, we plan to fabricate an actual hemispherical resonator and conduct performance tests to validate the accuracy of the multi-objective optimization results presented in this study.

## Figures and Tables

**Figure 1 micromachines-16-00287-f001:**
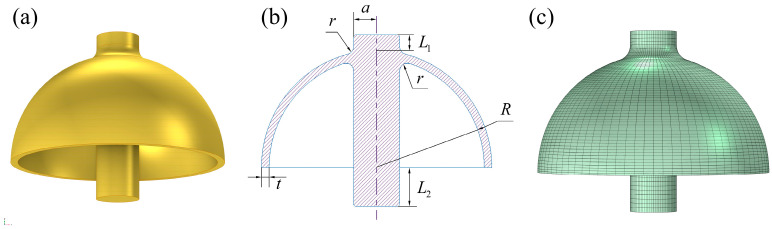
Schematic structure and uniform finite element meshing of hemispherical resonator. (**a**) physical diagram. (**b**) geometric structure schematic. (**c**) uniform division of the hemispherical resonator mesh.

**Figure 2 micromachines-16-00287-f002:**
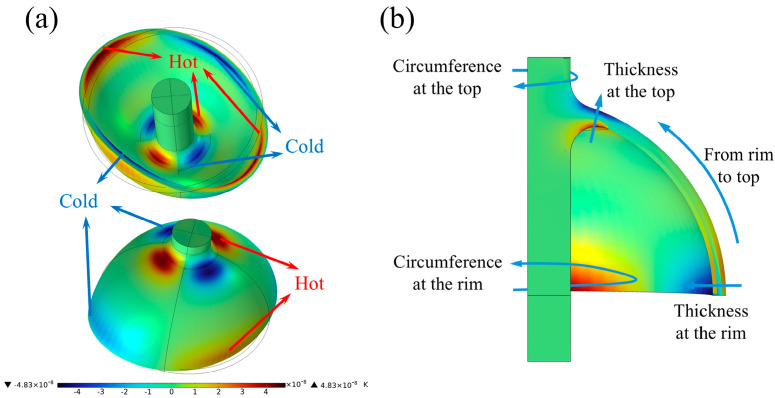
Temperature field distribution and heat conduction path in the operating mode of hemispherical resonator. (**a**) Temperature field distribution in operating mode. (**b**) Heat transfer paths in operating modes.

**Figure 3 micromachines-16-00287-f003:**
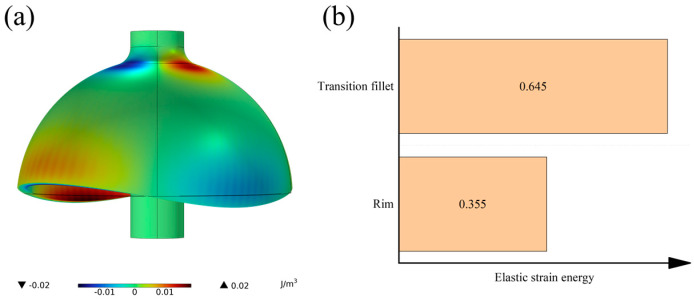
Plot of elastic strain energy analysis of hemispherical resonator. (**a**) Cloud diagram of elastic strain energy distribution. (**b**) Elastic strain energy distribution ratio.

**Figure 4 micromachines-16-00287-f004:**
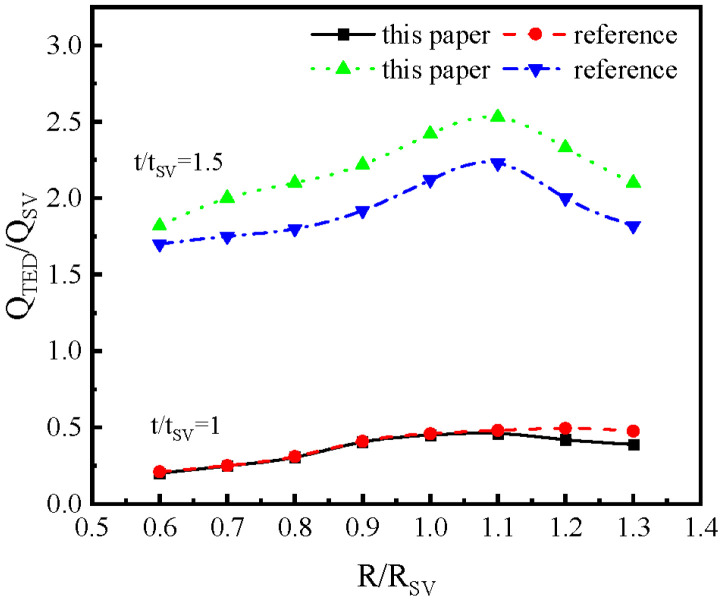
Comparison of the finite element model presented in this paper with the model from the literature [21].

**Figure 5 micromachines-16-00287-f005:**
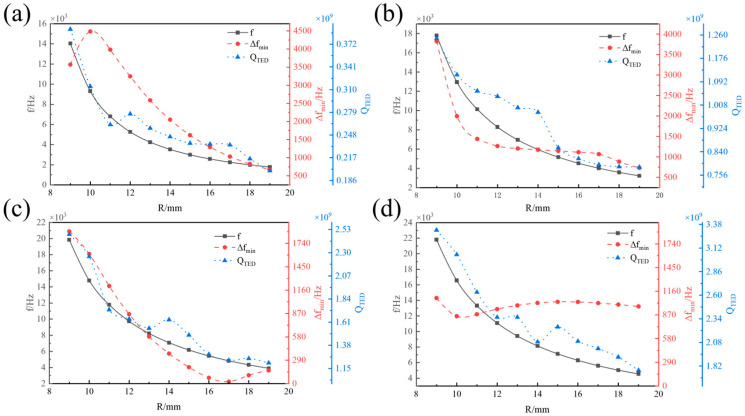
Influence of *R* on *f*, ∆*f_min_*, and *Q_TED_*. (**a**) *t* = 0.5 mm. (**b**) *t* = 1 mm. (**c**) *t* = 1.25 mm. (**d**) *t* = 1.5 mm.

**Figure 6 micromachines-16-00287-f006:**
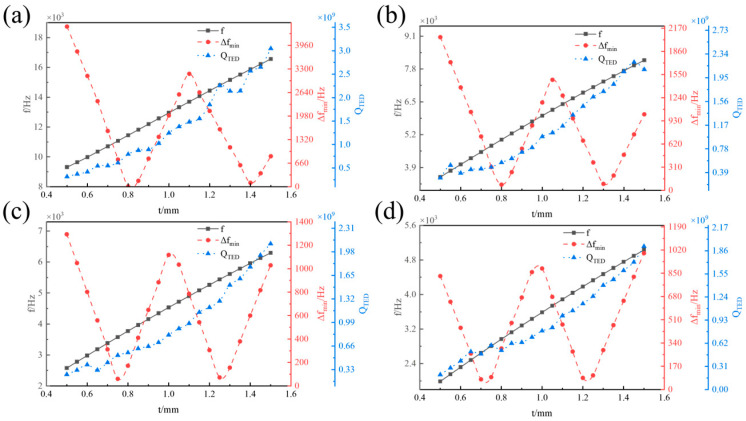
Influence of *t* on *f*, ∆*f_min_*, and *Q_TED_*. (**a**) *R* = 10 mm. (**b**) *R* = 14 mm. (**c**) *R* = 16 mm. (**d**) *R* = 18 mm.

**Figure 7 micromachines-16-00287-f007:**
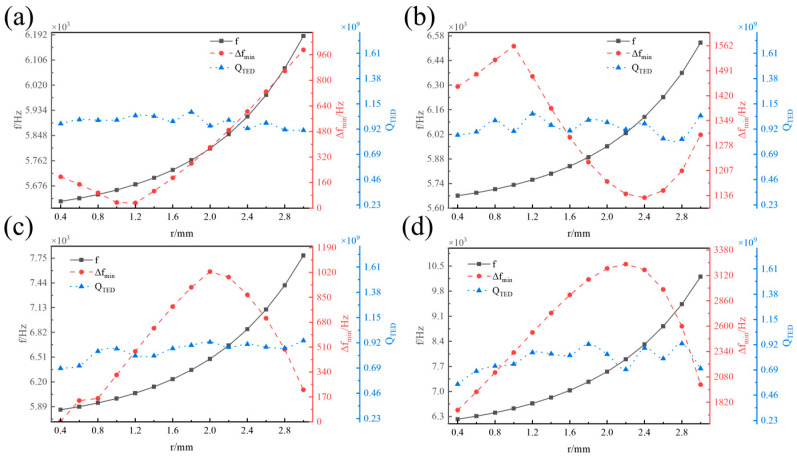
Influence of *r* on *f*, ∆*f_min_*, and *Q_TED_*. (**a**) *a* = 2.5 mm. (**b**) *a* = 3 mm. (**c**) *a* = 4 mm. (**d**) *a* = 5 mm.

**Figure 8 micromachines-16-00287-f008:**
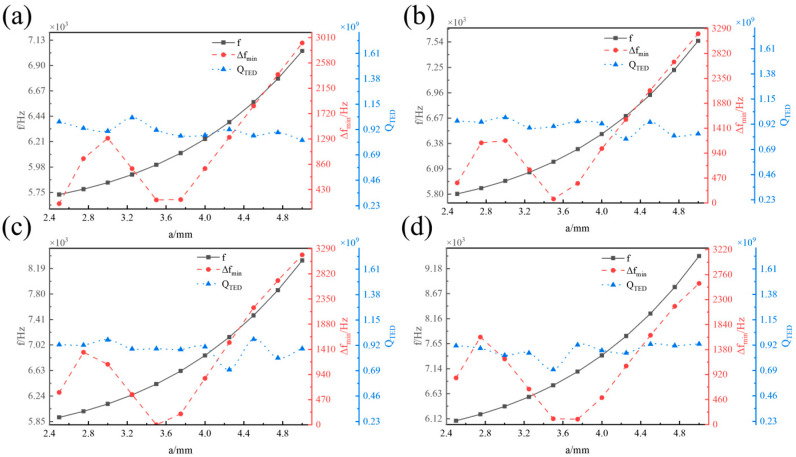
Influence of *a* on *f*, ∆*f_min_*, and *Q_TED_*. (**a**) *r* = 1.6 mm. (**b**) *r* = 2 mm. (**c**) *r* = 2.4 mm. (**d**) *r* = 2.8 mm.

**Figure 9 micromachines-16-00287-f009:**
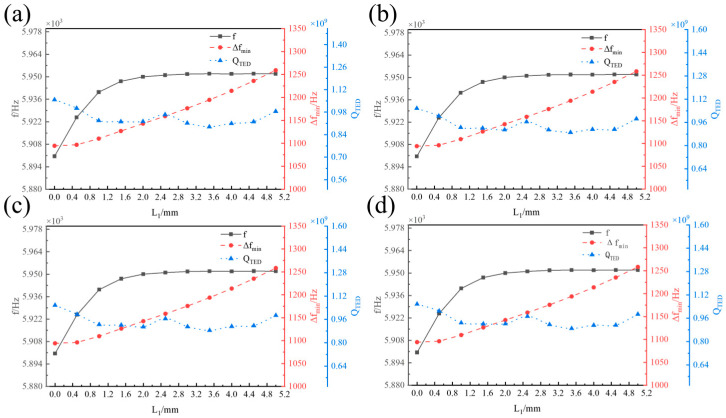
Influence of *L*_1_ on *f*, ∆*f_min_*, and *Q_TED_*. (**a**) *L*_2_ = 3 mm. (**b**) *L*_2_ = 5 mm. (**c**) *L*_2_ = 8 mm. (**d**) *L*_2_ = 10 mm.

**Figure 10 micromachines-16-00287-f010:**
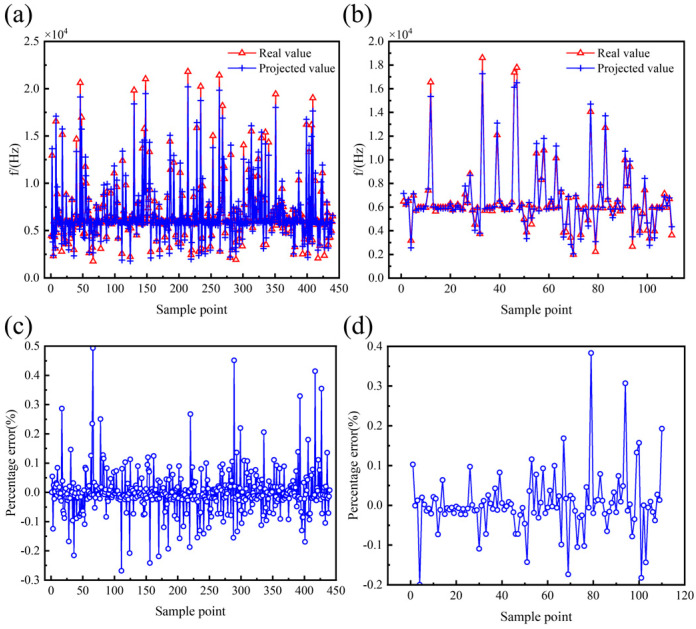
Effect of multivariate nonlinear regression model of *f*. (**a**) Comparison of true and predicted values of training set. (**b**) Comparison of true and predicted values for testing set. (**c**) Percentage error between true and predicted values of training set. (**d**) Percentage error between true and predicted values of testing set.

**Figure 11 micromachines-16-00287-f011:**
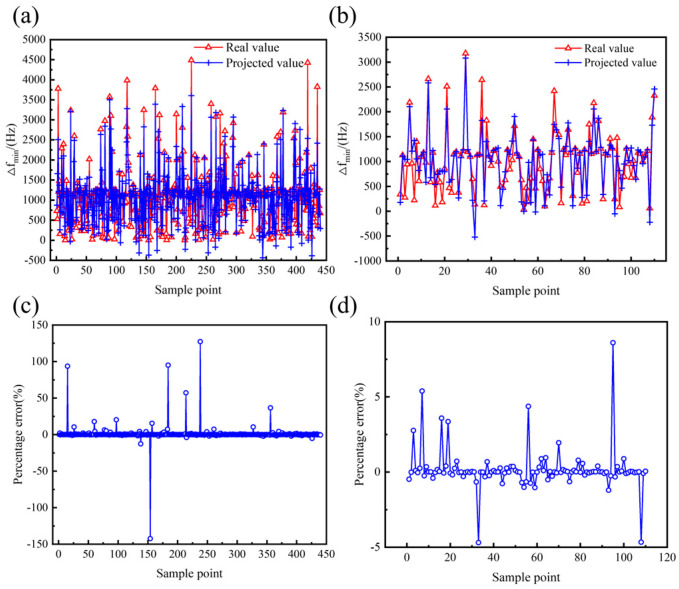
Effect of multivariate nonlinear regression model of ∆*f_min_*. (**a**) Comparison of true and predicted values of training set. (**b**) Comparison of true and predicted values for testing set. (**c**) Percentage error between true and predicted values of training set. (**d**) Percentage error between true and predicted values of testing set.

**Figure 12 micromachines-16-00287-f012:**
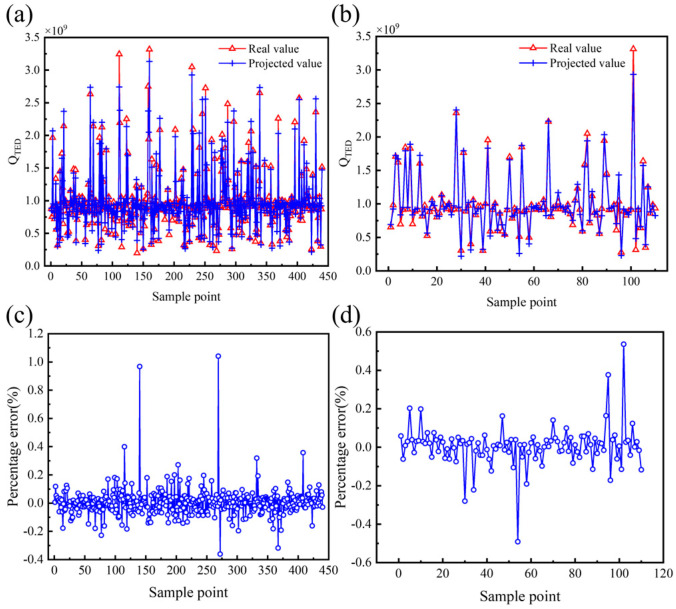
Effect of multivariate nonlinear regression model of *Q_TED_*. (**a**) Comparison of true and predicted values of training set. (**b**) Comparison of true and predicted values for testing set. (**c**) Percentage error between true and predicted values of training set. (**d**) Percentage error between true and predicted values of testing set.

**Figure 13 micromachines-16-00287-f013:**
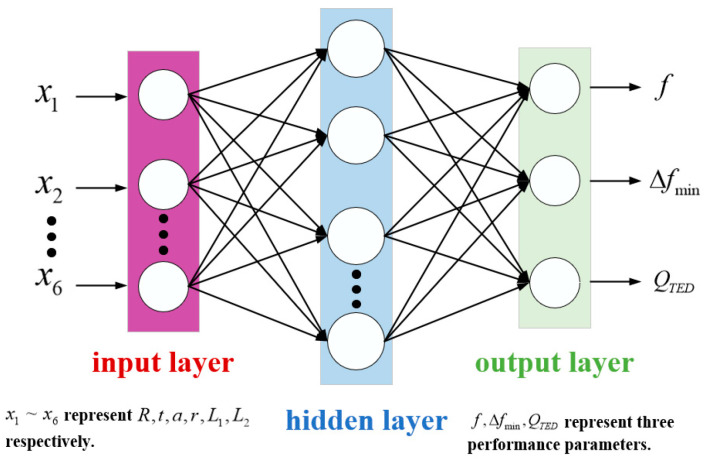
Structure of BP neural network.

**Figure 14 micromachines-16-00287-f014:**
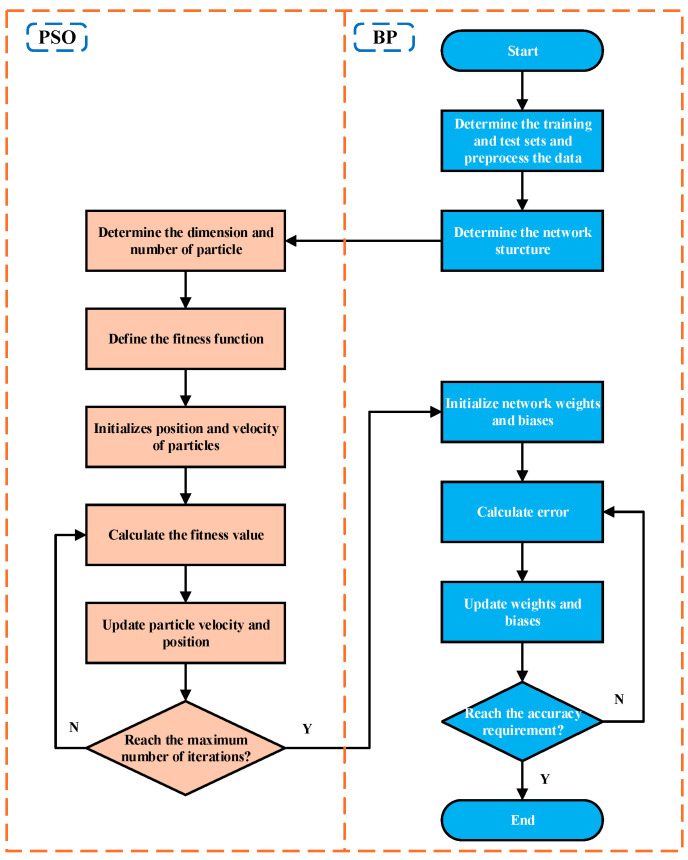
Structure of PSO-BP neural network.

**Figure 15 micromachines-16-00287-f015:**
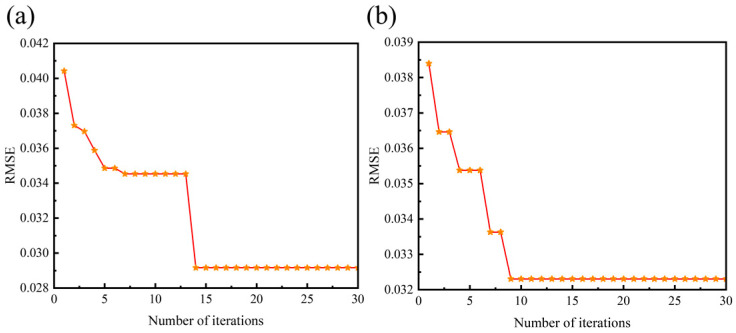
Particle swarm optimal fitness change curve. (**a**) Phase 1. (**b**) Phase 2.

**Figure 16 micromachines-16-00287-f016:**
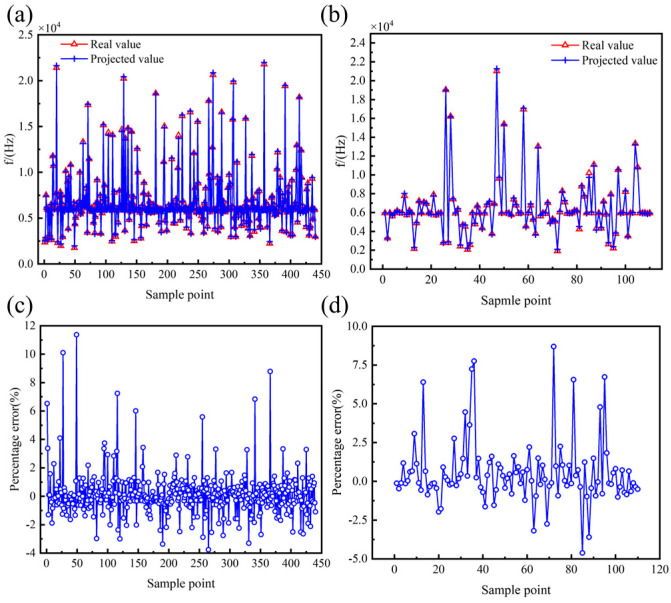
Effectiveness of the PSO-BP prediction model of *f*. (**a**) Comparison of true and predicted values of the training set. (**b**) Comparison of true and predicted values for the testing set. (**c**) Percentage error between true and predicted values of the training set. (**d**) Percentage error between true and predicted values of the testing set.

**Figure 17 micromachines-16-00287-f017:**
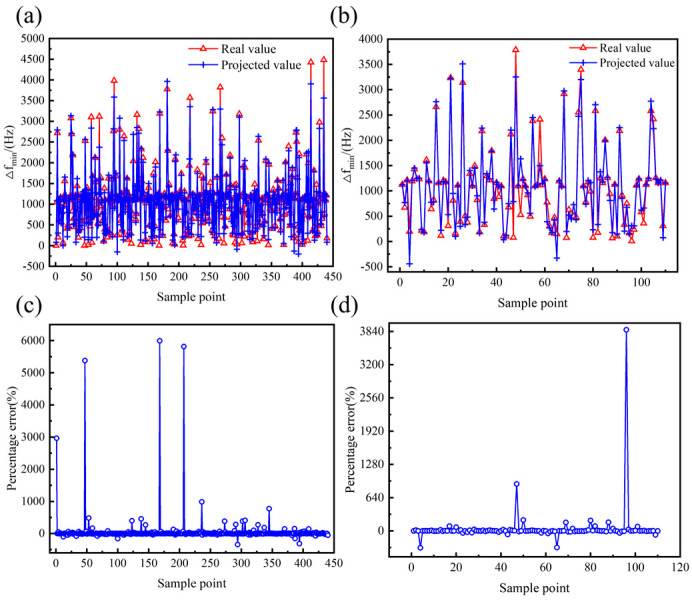
Effectiveness of the PSO-BP prediction model of ∆*f_min_*. (**a**) Comparison of true and predicted values of the training set. (**b**) Comparison of true and predicted values for the testing set. (**c**) Percentage error between true and predicted values of the training set. (**d**) Percentage error between true and predicted values of the testing set.

**Figure 18 micromachines-16-00287-f018:**
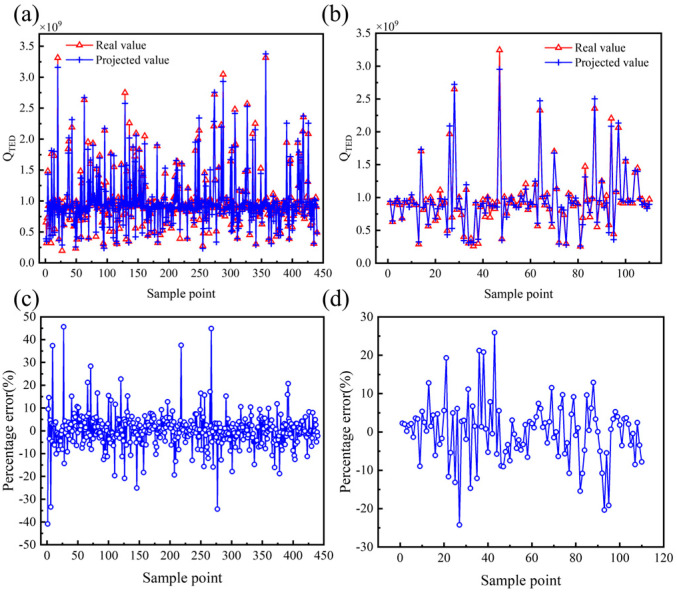
Effectiveness of PSO-BP prediction model of *Q_TED_*. (**a**) Comparison of true and predicted values of the training set. (**b**) Comparison of true and predicted values for the testing set. (**c**) Percentage error between true and predicted values of the training set. (**d**) Percentage error between true and predicted values of the testing set.

**Figure 19 micromachines-16-00287-f019:**
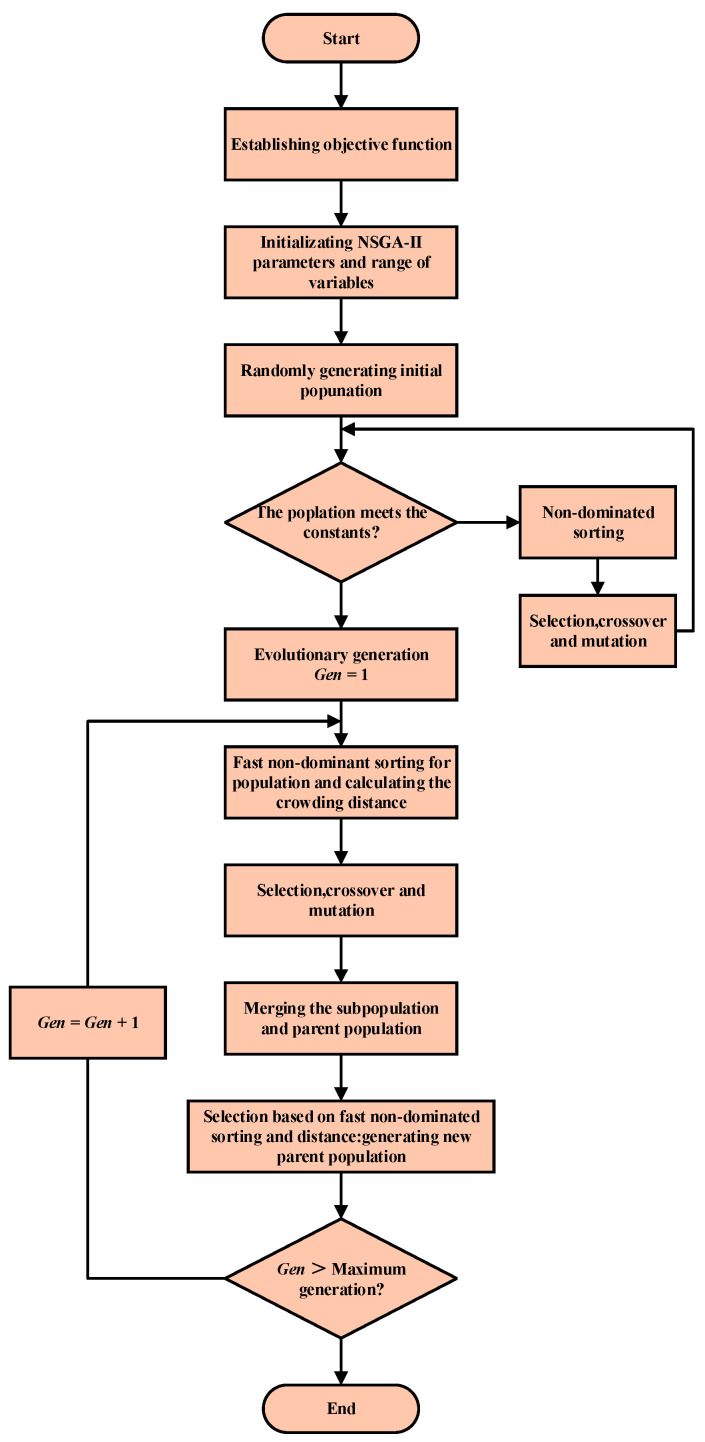
Flowchart of NSGA-II algorithm.

**Figure 20 micromachines-16-00287-f020:**
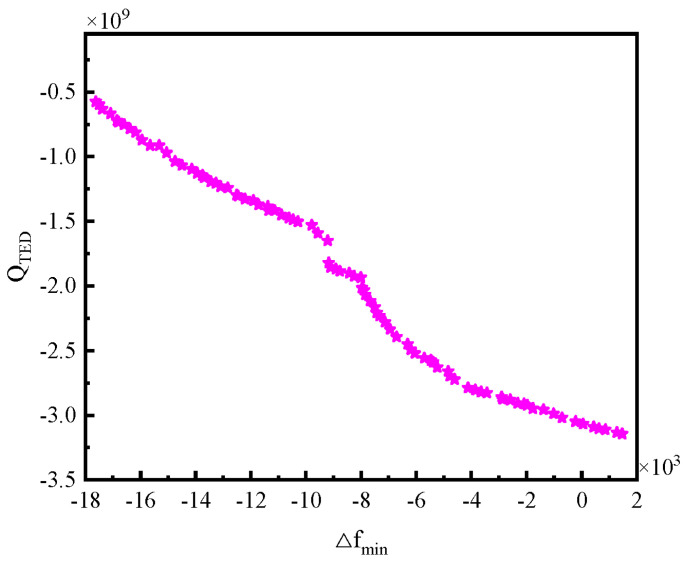
Pareto solution set for NSGA-II algorithm.

**Table 1 micromachines-16-00287-t001:** Initial structural parameters of hemispherical resonator.

Parameter Name	Symbols	Sizes (mm)
midplane radius	*R*	10
shell thickness	*t*	1
centre pole radius	*a*	3
inner and outer corner radius	*r*	2
length of centre bar clamping end	*L* _1_	3
length of fixed end of centre rod	*L* _2_	5

**Table 2 micromachines-16-00287-t002:** Fused silica material parameters.

Parameters	Value
modulus of elasticity *E* (GPa)	76.7
density *ρ* (kg/m^3^)	2200
Poisson’s ratio *μ*	0.17
thermal conductivity *k* (W/(m·K))	1.38
coefficient of linear expansion *α* (1/K)	5.5 × 10^−7^
specific heat capacity *C* (J/(kg·K))	740

**Table 3 micromachines-16-00287-t003:** Frequency cleavage values of hemispherical resonator for different meshing methods.

Grid Division Method	Operating Frequency 1/Hz	Operating Frequency 2/Hz	Frequency Cracking Value/Hz
Hyperfine free hexahedral mesh	12,948.046576	12,948.656119	0.609543
Extremely fine free hexahedral meshes	12,948.196519	12,948.298621	0.102102
Uniform segmentation of structured grids	12,950.336465	12,950.336564	0.000099

**Table 4 micromachines-16-00287-t004:** Hemispherical resonator geometry parameter setting table.

Parameter Name	Basic Value	Parameter Range	Simulation Step
*R*/mm	14	9–19	1
*t*/mm	1	0.5–1.5	0.05
*a*/mm	3	2.5–5	0.25
*r*/mm	2	1–3	0.2
*L*_1_/mm	3	0–5	0.5
*L*_2_/mm	5	3–10	0.5

**Table 5 micromachines-16-00287-t005:** Analysis results of multiple linear regression models.

Performance Parameters	Training Set			Testing Set		
	*R* ^2^	MAE	RMSE	*R* ^2^	MAE	RMSE
*f*	0.81366	936.2167	1365.772	0.79214	873.3526	1293.087
∆*f_min_*	0.34065	443.6096	620.208	0.22183	399.0484	545.9517
*Q_TED_*	0.86462	1.112 × 10^8^	1.697 × 10^8^	0.86646	1.118 × 10^8^	1.675 × 10^8^

**Table 6 micromachines-16-00287-t006:** Analytical results of multiple nonlinear regression models.

Performance Parameters	Training Set			Testing Set		
	*R* ^2^	MAE	RMSE	*R* ^2^	MAE	RMSE
*f*	0.97903	297.7542	458.1642	0.97613	282.5439	438.1888
∆*f_min_*	0.82137	193.9606	322.0969	0.77184	190.281	295.6247
*Q_TED_*	0.97302	5.121 × 10^7^	7.575 × 10^7^	0.96948	5.263 × 10^7^	8.006 × 10^7^

**Table 7 micromachines-16-00287-t007:** Analytical results of PSO-BP regression models.

Predictive Model	Performance Parameters	Training Set			Testing Set		
		*R* ^2^	MAE	RMSE	*R* ^2^	MAE	RMSE
*BP*	*f*	0.99482	155.958	211.4748	0.99578	169.4341	239.5641
	∆*f_min_*	0.94215	103.5494	170.0201	0.91359	141.7758	247.4418
	*Q_TED_*	0.97303	5.1417 × 10^7^	7.1832 × 10^7^	0.96763	6.7855 × 10^7^	9.7721 × 10^7^
*PSO-BP*	*f*	0.99798	93.3561	134.5081	0.99741	114.4461	187.6712
	∆*f_min_*	0.96777	80.6316	126.9067	0.96145	108.1149	165.279
	*Q_TED_*	0.98215	4.0774 × 10^7^	5.8448 × 10^7^	0.97335	6.227 × 10^9^	8.8666 × 10^7^

**Table 8 micromachines-16-00287-t008:** Comparison of performance parameters before and after optimization.

	Projected Value		Actual Value	
	∆*f_min_*	*Q* _TED_	∆*f_min_*	*Q* _TED_
Pre-optimization	2005.691	1.162 × 10^9^	1996.513	1.247 × 10^9^
Post-optimization	2142.834	2.149 × 10^9^	2088.546	2.125 × 10^9^

## Data Availability

The original contributions presented in this study are included in the article. Further inquiries can be directed to the corresponding author(s).

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
