# Peer review of "Comprehensive Performance-Oriented Multi-Objective Optimization of Hemispherical Resonator Structural Parameters"

_micromachines, 2025, doi:10.3390/mi16030287_

Round 1

Reviewer 1 Report

Comments and Suggestions for Authors

In the present study, the authors describe a procedure to infer the relationship between the geometric features to the performance of hemispherical resonant gyroscopes. The problem is approached through a model developed by means of a multi-physics software, followed by the application of regression models and optimization algorithms to identify the most effective configurations. Given the wide range of applications for these resonators, the manuscript is certainly relevant; however, some clarifications would enhance its quality.

  1. A key concern is the symmetry of the structure, which the authors acknowledge in their conclusions. Since real devices often exhibit imperfections (e.g., geometric variations, non-homogeneous material properties), this may limit the findings to a qualitative analysis. A significant contribution would involve validating the inferred laws with experimental results. Currently, the manuscript appears to primarily apply known regression and optimization algorithms to a test case.

Here are a few additional suggestions for improvement:

  1. When studying symmetric structures, it is important to use a symmetric mesh to minimize the risk of spurious numerical solutions. The brought novelty form the “multi-zone” approach are already well-know in the literature.
  2. In Table 2, "Poisson’s ratio" should be correctly capitalized.
  3. Line 145 : "n"—could the authors please specify what this represents?
  4. Line 147: the discussion regarding the 45° angular shift between modes would benefit from further elaboration. It may be helpful to explore the symmetry of these devices, particularly concerning the existence of “conjugate” modes with specific angular shifts (see for example “Kubenko, V.D., Koval’chuk, P.S., Krasnopol’skaya, T.S.:Effect of initial camber on natural nonlinear vibrations of cylindrical shells. Sov. Appl. Mech. 18(1), 34–39 (1982).”).
  5. Line 187: for clarity, it would be helpful to define QTED before using the acronym.
  6. Increasing the resolution of all figures is recommended, particularly for Figures 7-11, as they currently may not meet publication standards.
  7. Reference [2] in the List of References appears incomplete.

Best regards

Reviewer 2 Report

Comments and Suggestions for Authors

   The paper “Comprehensive Performance-Oriented Multi-Objective Optimization Of Hemispherical Resonator Structural Parameters” proposes a method for optimizing the design parameters of HRG hemispherical resonators. As the object of the study, the authors chose the resonator with the hemisphere radius of 10 mm and the wall thickness of 1 mm. Using finite element simulation the authors determined the influence of the resonator design parameters on thermoelastic loss and the minimum frequency difference. Then, using the proposed mathematical model, the design parameters of the resonator were optimized to obtain the best values ​​of Qted and dFmin. The design parameters of the resonator, optimized according to these criteria, are presented. In the resonator with the optimal design, the dFmin is improved by 6.97%, and in the Qted – by 128%. The advantage of the paper is the combination of the analysis of the influence of design parameters on the characteristics of the resonator and the model for optimizing these parameters.

   The most significant drawback of the paper is the unsuccessful choice of the object of study, since for resonators of such dimensions, thermoelastic internal friction is very small. According to the authors’ calculations, the quality factor of the resonator, limited by thermoelastic loss, is approximately E9, while the quality factor of real resonators of this type is approximately E7, that is, two orders of magnitude less. Therefore, optimization of the resonator design parameters by the value of the Qted is ineffective, since in practice the gain in the resonator quality factor will be very small. On the other hand, during the optimization the authors did not take into account a number of other factors that are important for the resonator. In section 5 Conclusion, the authors present the main dimensions of the optimal (according to the obtained solution) resonator: radius - 9.083 mm, wall thickness - 1.497 mm, stem radius - 2.836 mm, transition radii -1.689 mm. Although the authors do not provide the value of the oscillation frequency for this resonator, based on the graphs in Fig. 7-10, the oscillation frequency will be 18-20 kHz. In this case, the Qted chosen as one of the criteria for optimization does increase. But in reality, the increase in the oscillation frequency simultaneously leads to an increase in internal friction in the fused silica, so the quality factor of the resonator operating at a frequency of 18-20 kHz will be much lower than the quality factor of the resonator operating at a frequency of 8-10 kHz. In addition, due to the increase in frequency, the damping time of oscillations in the resonator will be significantly reduced, and this is one of the most important parameters of the resonator. Also, in a resonator with a relatively thick shell wall, nonlinear effects will be more pronounced. In addition, the high oscillation frequency will require a significant increase in the speed of electronic control systems. From the reviewer's point of view, the design of the resonator obtained by the authors using the proposed method is unsuccessful, which does not allow us to evaluate the reliability and effectiveness of this method.

   This result of the optimization is due to the fact that the authors optimized the resonator design according to a secondary (for this resonator) parameter.

The influence of thermoelastic loss becomes significant in microresonators with a small radius and thin wall (for example, in the resonator described in [D.Senkal et al. “1 million Q-factor demonstrated on micro-glassblown fused silica wineglass resonators with out-of-plane electrostatic transduction”//Solid-State Sensors, Actuators and Microsystems. Workshop. Hilton Head Island, South Carolina, June 8-12, 2014. P.68-71] or others of similar dimensions). The method proposed by the authors would most likely be suitable for optimizing the design of such microresonators. However, applying this method to a large-sized resonator would rather discredit the method proposed by the authors than demonstrate its merits.

   From the reviewer's point of view, the paper requires major revision. It is necessary to use another object of study and conduct similar calculations for some microresonator, the design of which is described in the literature, then the possibilities and advantages of the proposed method will be obvious.

Round 2

Reviewer 1 Report

Comments and Suggestions for Authors

N.A.

Author Response

Thank you for your review!

Reviewer 2 Report

Comments and Suggestions for Authors

The changes made by the authors improve the manuscript, but they are not related to my main comment: the authors chose the unsuccessful criterion for optimizing (minimum thermoelastic losses) the resonator design and as a result they obtained the unsuccessful resonator design.

There are various types of vibration energy losses in the resonator, such as anchor fastening, surface loss. However, they are associated with defects that occur during the manufacture of the resonator, i.e., they are associated with the production technology, and not with the design of the resonator. Air damping loss are also associated with the HRG vacuum technology.

When constructing the resonator itself, one of the main purposes is to achieve the maximum oscillation damping time; to do this, it is necessary to obtain the highest possible quality factor at a moderately high oscillation frequency. Thermoelastic loss, of course, affect the quality factor of the resonator, but in this case (for resonators of a relatively large size), the main factor is the dependence of internal friction in the fused quartz on the oscillation frequency.

With the increase of the oscillation frequency, the internal friction in fused quartz increases significantly (this is a fundamental phenomenon that cannot be avoided), and thermoelastic loss, on the contrary, decreases. In the resonator design obtained by the authors, the thermoelastic loss is indeed small, but the bulk internal friction in fused quartz will be large and, in general, the quality factor of the resonator (and especially the damping time of oscillations in the resonator) will be comparatively small.

That is, the optimization of the resonator design proposed by the authors led to the unsuccessful solution. The obtained result does not confirm the effectiveness of the developed optimization method, it rather discredits it. However, for microresonators, this approach is quite justified, because in such resonators, thermoelastic loss is the dominant factor. Therefore, my proposal is still that the authors apply this method to optimize some microresonator design where it is most likely to give a good result, and on this basis, rework the article.

Round 3

Reviewer 2 Report

Comments and Suggestions for Authors

The corrections made really improved the paper. There are a few minor comments.

1.The content of the following fragment (lines 43-49) is not entirely correct:

“A higher vibration frequency enhances the sensitivity of the hemispherical resonator gyro (HRG); however, it also increases power consumption and manufacturing complexity. High‐frequency vibration necessitates advanced manufacturing techniques to maintain the uniformity and symmetry of the hemispherical shell. Furthermore, the drive and detection circuits must meet more stringent requirements. The vibration frequency of aerospace‐grade hemispherical resonator gyros (HRGs) typically ranges from 4 to 8 kHz.”

I would recommend rewriting it, for example, as follows:

“Achieving high accuracy of the HRG requires a long time for the vibrations to decay in the resonator. For this, the resonator's quality factor must be as high as possible, and the vibration frequency must be low. However, to reduce the vibration frequency, it is necessary to reduce the thickness of the resonator wall, which complicates the production of resonators and can disrupt the axial symmetry of the resonator. That’s why the vibration frequency of aero‐space‐grade HRGs typically ranges from 4 to 8 kHz.”

2. Fig. 1 shows the general appearance of the resonator and its main dimensions. Fig. 2 shows the division of the resonator into finite elements. To reduce the volume of the paper, these figures can be combined, for example, by showing the finite element mesh in Fig. 1.

3. Fig. 3 has been published in various papers many times. There is no need to repeat it to show the bending of the shell, especially since the bending of the resonator shell is also shown in Fig.4. To reduce the volume of the paper, Fig. 3 can be excluded.

4. In References:  all authors should be cited in the reference [1].

After some minor revision, this paper can be published.
